# Variations in Postmenopausal Body Composition: A Cross-Sectional Comparison between Physical Activity Practitioners and Sedentary Individuals

**DOI:** 10.3390/jfmk9010012

**Published:** 2023-12-30

**Authors:** Camila Mahara Dias Damasceno, Fernando José de Sá Pereira Guimarães, Keyla Brandão Costa, Ana Claudia Morais Godoy Figueiredo, Rodrigo Cappato de Araújo, Manoel da Cunha Costa

**Affiliations:** 1Graduate Program in Rehabilitation and Functional Performance, University of Pernambuco, BR 203, Km 2 s/n–Vila Eduardo, Petrolina CEP 56328-900, PE, Brazil; fernando.guimaraes@fmo.edu.br (F.J.d.S.P.G.); rodrigo.cappato@upe.br (R.C.d.A.); manoel.costa@upe.br (M.d.C.C.); 2Higher School of Physical Education, University of Pernambuco, Arnóbio Marques Street, 310, Santo Amaro, Recife CEP 50100-130, PE, Brazil; keyla.costa@upe.br; 3Master’s Program in Health Sciences at the Foundation for Teaching and Research in Health Sciences (FEPECS), North Hospital Medical Sector—Asa Norte, Brasília CEP 70710-907, DF, Brazil; contato@anacgodoy.com.br

**Keywords:** female health, menopausal transition, sedentary behavior, exercise engagement, physiological adaptations, lifestyle comparison

## Abstract

Physical activity is broadly recognized for promoting weight reduction and bestowing numerous health benefits. Nonetheless, robust evidence concerning the impact of physical activity on postmenopausal women, undergoing physiological shifts, remains scant. This study aimed to elucidate the relationship between physical activity and body composition among postmenopausal women. Employing a cross-sectional and retrospective design, 702 women were examined. Data on physical activity and body composition were amassed through anthropometric assessments and Dual-Energy X-ray Absorptiometry (DEXA). A significant proportion of women exhibited anthropometric alterations indicative of overweight/obesity, alongside elevated values in Waist Circumference (WC), Waist-to-Hip Ratio, Waist-to-Height, and bone mass, signifying a heightened risk for disease onset. While a majority engaged in some form of physical activity, this did not yield notable reductions in the assessed metrics. Noteworthy changes were only discerned in BMI and bone mass among pre-menopausal women; whereas among postmenopausal women, in addition to disparities in bone mass, those inactive were 1.18 times more prone to a very high disease risk, as gauged by WC.

## 1. Introduction

Menopause results from the cessation of ovarian function, marking the end of menstrual cycles, typically around the age of 50. It is deemed premature when occurring before 40, and precocious if between the ages of 40 and 45 [1]. Postmenopause, the phase following the cessation of a woman’s reproductive ability, is bifurcated into early (lasting 5 to 8 years) and late stages, extending until death. The early stage encapsulates hormonal shifts such as a continuous rise in Follicle-Stimulating Gonadotropin Hormone (FSH) and a decline in estradiol [2].

Under the sway of diminishing estrogen levels during this phase, women undergo myriad alterations including weight fluctuation, fat redistribution, and body composition changes—marked by increased fat, reduced muscle mass, and bone tissue—with direct ramifications on female health [3]. These shifts are earmarked as risk factors for disease onset [4]. When menopause and its sequela manifest prematurely, additional risks surface [5,6].

Body Composition (BC) gauges the primary constituents of the human body, segmenting it into two to four component models (bone, muscle, fat, visceral, or other tissues), assessed through laboratory and clinical methods. This measurement facilitates disease diagnosis and intervention monitoring, such as physical activity, with gender, age, genetic, and environmental factors influencing body mass variations among individuals [7,8].

Various tools and complex procedures like Absorptiometry or Dual-Energy X-ray Densitometry (DEXA) are employed to measure Waist Circumference (WC), with DEXA being a reference technique for evaluating three essential components for human health and performance monitoring: bones, muscles, and body fat [9].

The literature generally underscores the manifold health benefits of physical activity, such as diminished disease risk and mortality from diverse causes [10]. It serves as an efficacious aid in weight management [11], thereby curtailing disease development driven by fat accumulation and/or muscle mass loss, and aiding in body composition amelioration. However, during menopause, the influence of physical activity may not remain consistent [12]. This period witnesses a decline in physical performance, tied to muscle mass and strength loss [13], potentially affecting exercise performance. Thus, this study endeavors to ascertain: Does physical activity significantly impact the body composition of postmenopausal women?

Considering the augmentation in life expectancy, where women increasingly endure menopause and its aftermath [13], more emphasis should be accorded to this demographic. Hence, research exploring the nexus between menopause and viable therapeutic avenues, like physical activity, merits attention to enhance these women’s quality of life and mitigate disease risk.

The existing scientific literature correlates risk factor control—such as smoking, obesity, dyslipidemias, and sedentary lifestyle—with disease prevention. Yet, concerning lifestyle interventions pertaining to menopause, extant studies exhibit lacunae [1]. The clarity on whether physical activity sufficiently and significantly mitigates the morphological alterations consequent to menopause, like muscle mass and bone tissue loss, remains elusive [12].

As of now, in postmenopausal women, the interplay between physical activity and body composition still harbors uncertainties. The plethora of variables introduced by physical activity, encompassing diverse exercises, durations, and intensities, complicates the assessment of its impact, even on disease development. Ergo, further investigation is warranted to evaluate the responses of this cohort to exercise regimens [14]. This study aims to examine the bearing of activity on body composition in postmenopausal women.

## 2. Materials and Methods

### 2.1. Study Design

A cross-sectional study was conducted. The study was approved by the Research Ethics Committee of the State University of Pernambuco (UPE), Recife, Brazil and registered under number 2,332,880.

### 2.2. Participants (Selection of Subjects, Inclusion Criteria, Exclusion, and Losses)

In this study, 702 pre- and post-menopausal women were included, aged between 18 and 87 years. Research participants were recruited by convenience after invitation via social media, posters, and call by the university. After data collection, considering a confidence level of 95% and alpha error of 5% and power of 80%, it was possible to verify that the sample of 702 women was sufficient to test the hypothesis investigated. The calculation was conducted in OpenEpi (https://www.openepi.com/SampleSize/SSPropor.htm (accessed on 1 December 2023)).

Women over the age of 18 were included, apparently healthy and who had no limitations in carrying out motor tests.

Pregnant women, women who tested positive for the Human Immunodeficiency Virus (HIV), undergoing oncological treatment, who were unable to undergo the DEXA exam, and who used medications that modified body composition or who presented an acute pathological condition that caused changes in body composition were excluded. 

The diagnosis of menopause was self-reported by the volunteers, who mentioned the date of the last menstruation.

### 2.3. Data Collection

After the presentation of the project, the interested parties went through an interview to identify the women eligible to participate in the research. After eligibility, sociodemographic and physical exercise information was collected. Anthropometric measurements, as well as DEXA imaging exams, were also collected.

Participants who mentioned regularly engaging in physical activity were considered active and those who did not systematically perform physical activity were considered sedentary.

### 2.4. Variables

The variables used were self-reported physical activity (<150 min = sedentary; ≥150 min = active), type of self-reported physical activity (weight training, aerobics, and others), Body Mass Index (18.5 kg/m^2^ to 24.9 kg/m^2^ = normal; 25 kg/m^2^ to 29.9 kg/m^2^ = overweight; ≥30 kg/m^2^); Bone mass (≤−1 standard deviations from the mean of the parameter = normal; −1.1 to −2.4 standard deviations = osteopenia and ≥−2.5 standard deviations from the mean = osteoporosis), Waist-to-height ratio (<0.5 = Low risk; ≥0.5 = High risk); Waist Circumference (<0.8 = Risk-free; 0.8–0.9 = High risk; ≥1.0 = Very high risk); Appendicular lean mass (<6.47 kg/m^2^ = Low; ≥6.47 kg/m^2^ = Discharge) and Fat Mass Index (FMI) (<4.6 kg/m^2^ = Normal; ≥4.6 kg/m^2^ and <5.7 kg/m^2^ = Overweight; ≥ 5.7 kg/m^2^ = Obesity).

### 2.5. Data Analysis

Mean and standard deviation were calculated for age. For the qualitative variables, the chi-square test or Fisher’s exact test was used to test the hypotheses, in addition to presenting absolute and relative values of the categories. In addition, unconditional logistic regressions were conducted to estimate the crude and adjusted prevalence ratios of the outcomes investigated. At first, the “logistic” command of Stata was adopted to generate a crude and adjusted Odds Ratio, and then the measurement was converted into a Prevalence Ratio (PR) in 95% Confidence Interval (CI), Lower Confidence Interval (LCI), and Upper Confidence Interval (UCI) for each category of the variable in relation to the reference category by “odds risk” command.

The adjustment variables included in the multiple models were only age, race, and skin color. These variables were chosen based on the scientific literature on the subject. There was an adjustment only for age, then only for race/color, and finally for age and race/color simultaneously. All analyses were performed using the STATA software, version 16, serial number: 301606315062.

## 3. Results

Most of the women were over 45 years of age (65.91%), did some type of physical activity, with 40.48% aerobic exercise, and had menopause after 45 years, characterized as physiological menopause. Only 3.91% of the women had menopause classified as premature or early. In this group of women, most were dun (45.55%) and the ratio of dun to white was 42.06%. Most performed some type of physical activity (52.16%). Aerobic exercise was performed by 42.06% of these women.

It was observed that most women have risk factors for the development of diseases, such as changes in the Body Mass Index (with more than 68% of women being overweight), Fat Mass Index (38.97% overweight and 40.59% obese), Waist Circumference (more than half presenting some degree of risk), Waist-Height Ratio (almost 70% at high risk), Waist-to-Hip Ratio (just over 74% with high risk), and more than 50% with changes in bone mass. Only the Appendicular Lean Mass index showed favorable results, with more than 84% of the women presenting a high index.

In this study, considering the *p*-value, the performance of physical activity among women did not interfere with any of the anthropometric measurements analyzed (Appendix A).

Considering only women who did not go through menopause, a statistically significant difference was identified in Body Mass Index (*p* = 0.048), and bone mass between active and non-active women, Table 1. Women who do not engage in physical activity are 1.49 times more likely to have osteopenia when compared to women who do practice physical activity.

Among women who had already gone through menopause, a statistically significant difference was identified in bone mass (0.027) between active and non-active women. For waist circumference, this group is 1.18 times more likely to have a very high risk among women who do not engage in physical activity (Table 2).

The postmenopausal women are divided in three groups: physiological menopausal, premature menopausal, and early postmenopausal women. Only the premature menopausal showed significant differences in WC (*p* = 0.032), Table 3.

Table 4 shows the prevalence ratio and confidence interval of all women included in this study. It can be seen that only the Waist Circumference represented a statistically significant difference. Thus, it is understood that for this variable, women at very high risk, according to waist circumference, adjusted in the regression model for age and age × race/color, are 1.12 times more likely to be at very high risk than the group of women who do not practice physical activity.

For women who did not go through menopause, osteopenia adjusted for age (PR = 1.52), race/color (PR = 1.36), and age vs. race/color (PR = 1.52) showed a significant difference, Table 4. The interpretation of the cited findings is that women who did not practice physical activity were 1.52 times more likely to have osteopenia when compared to those who did, after adjusting for age and race/color. In addition, females in the absence of physical activity were 36% more likely to have osteopenia than in the comparison group, after adjusting for race/skin color alone. Therefore, women who do not practice physical activity have up to a 52% probability of having osteopenia compared to the comparison group.

Table 4 shows the prevalence ratio of women who have already gone through menopause. The very high risk for Waist Circumference adjusted for age (PR = 1.29), race/color (PR = 1.28), race/color × age (PR = 1.29), and Waist Height Ratio by race/color (PR = 1.64), and race/color × age (PR = 1.65). Therefore, the very high risk for waist circumference may be at least 1.28 times higher in the group of women who do not practice physical activity compared to the opposite group. For the Waist Ratio Height, high risk can occur up to 1.65 times among women who do not perform physical activity. The other results of this study are presented in Appendix A.

## 4. Discussion

Most of the women included in this study are over 45 years of age and have alterations in anthropometric measurements, which is worrisome, as this is associated with higher morbidity and mortality due to diseases, such as cardiovascular diseases. If we consider that in menopause this risk increases, due to hormonal changes, care should be even greater.

For middle-aged women, physiological changes associated with the menopausal transition and aging process can make them more prone to weight gain, as well as having greater difficulty in losing the kilograms already gained, thus increasing the risk of disease.

Studies show that other risk factors such as poor diet, decreased physical activity, and even sleep interfere with weight gain and become even more important during the transition to menopause [15].

In this study, it was also observed that a high number of women adhered to the practice of physical activity. Most of them performed some modality, with aerobics as their main one. However, the performance of physical activity among these women did not show significant results in reducing the measurements. Significant changes were identified only in BMI and bone mass in women who had not gone through menopause, and postmenopausal women; in addition to differences found in bone mass, those who did not perform physical activity were 1.18 times more likely to have very high risk, considering WC.

Constancy in physical activity, in addition to improving quality of life, enables body fat control and prevention of chronic events such as cardiovascular, diabetes, and even osteoporosis [10]. Studies show that decreased physical activity interferes with weight gain and becomes even more important during the transition to menopause [15].

Physical activity is recognized to provide health benefits [16]. Safely, it can even reduce cardiovascular risk factors, so this healthy lifestyle habit should be encouraged [17].

However, considering postmenopausal women, the literature still presents controversial results regarding the interference of exercise in this group in some variables [18]. However, the numerous possibilities of existing exercises, such as resistance, low volume, and high intensity, allow the development of further studies to better understand their role in the health–disease–care process [17] and even favor doubts about physical activity in the context of women’s health.

Data show that depending on the level of physical activity, such as low-intensity exercises, the results may be less effective in relation to changes resulting from advancing age. Thus, longer and more intense training periods may be necessary for postmenopausal women [19].

These findings complement previously published information [20]. The authors evaluated 1539 overweight/obese men and women (mean age 65.3 years) and concluded that an increase of 1 h per day of moderate to intense physical activity was able to reduce BMI, WC, and fat mass while improving bone mass and muscle strength parameters. In this study, light physical activity did not show any statistically significant difference.

A cohort of middle-aged adults also identified that physical activity, when performed more intensely, contributes significantly to reducing the risk of cardiovascular disease. The intensity of physical activity seems to reduce the association, previously presented, between volume (intensity × time) and cardiovascular diseases, possibly from greater cardiorespiratory and vascular adaptation [21]. Thus, high-intensity activities should be encouraged, even in activities of daily living, such as walking.

Other research [19] argues that physical activity can positively influence the expected effects of aging. The authors mention that regular physical activity preserves vascular health, as it can reproduce some of the effects of endogenous estrogen, a hormone that reduces during menopause.

A cross-sectional study with postmenopausal women found that physical activity reduced fat mass, BMI, and WC, increasing lean mass. The authors stated that a moderate level of leisure-time physical activity can provide benefits, and should therefore be encouraged [22]. Other authors also present in their results that resistance exercise can positively interfere with muscle mass, fat mass, as well as bone tissue [23].

In summary, it can be seen that even without the current study presenting significant statistical results regarding the influence of physical activity on women’s body composition, other studies point to the power of physical activity in providing many benefits with clinical relevance. There is then a need to better explore the different exercise modalities to more accurately define the power of physical activity among women.

It is also known that the reports on physical activity are commonly biased [18]; in addition, with aging, physical activity tends to decrease [22] and since in this study, we considered only the description of the participants, the overestimation of the practice of physical activity can be considered a bias and therefore not many changes in body measurements were identified.

Addressing the strengths of this study, it is worth mentioning that the researchers were trained to develop the work, using a specialized laboratory, and validated tools to measure all body measurements, in addition to double-checking protocols. Everything was performed to enhance the best quality of information and reduce the likelihood of distortion of the results. To minimize confounding bias, multiple analysis was performed; however, there is still a possibility of residual confounding.

Finally, this study demonstrated the need for further investigation of the influence of physical activity on body composition among women. It is observed that greater attention should be paid to the different modalities of physical activity that exist, as well as to the different levels required to promote body changes depending on age.

## 5. Conclusions

Drawing from the findings delineated and the literature reviewed, it is deduced that a deeper investigation into the impact of physical activity on women’s body composition is warranted. Despite the lack of significant outcomes in the current study regarding the diminution of physical activity measures, it is pivotal to acknowledge that diverse exercise modalities and intensity levels can elicit varying effects on body composition, markedly across different age brackets. The results that are presented in this manuscript indications there is a need for deeper studys to state a strong conclusion. In addition, other longitudinal studies with a similar population are suggested in an attempt to elucidate answers on the subject that have not been exhausted in this study.

Middle-aged women, especially those traversing the menopausal transition, are prone to encountering physiological alterations that can modulate body composition, such as augmented weight gain and heightened challenges in weight loss.

In this vein, regular physical activity can serve as a linchpin in thwarting chronic ailments like cardiovascular disease, diabetes, and osteoporosis, in addition to ameliorating the quality of life.

Nonetheless, it is imperative to bear in mind that the outcomes of physical activity on body composition can fluctuate based on the intensity and the nature of the exercise undertaken. Research accentuates that resistance training, high-intensity exercises, and low-intensity exercises can harbor disparate impacts on body composition.

Hence, it is requisite to delve deeper into the different exercise modalities to unravel the potential of physical activity among women more lucidly. Furthermore, recognizing the limitations of the present study, such as recall bias and the absence of adequate sample calculation, is crucial. To propel this research domain forward, ensuing studies should encompass larger samples, more precise assessment methodologies, and strategies that curtail potential biases, like overestimation of physical activity engagement and residual confounding bias.

In summation, the comprehension of the effects of physical activity on the body composition of middle-aged and postmenopausal women remains in its infancy and beckons further exploration. It is indispensable to evaluate different exercise modalities and intensity levels, considering the unique characteristics of women at this life juncture.

Future investigations may shed light on the effects of physical activity on body composition, furnishing more precise and tailored counsel for health augmentation and disease deterrence in this demographic.

## Figures and Tables

**Table 1 jfmk-09-00012-t001:** Body indices derived from anthropometric and DEXA measurements of women who have not gone through menopause.

Variable	Physical Activity		*p* Value	PR (CI 95% LCI; UCI)
	Active N (%)	Sedentary N (%)		
Body mass index (BMI)				
Normal	82 (36.61)	76 (42.94)	0.048	0.73 (0.55; 0.97)
Overweight	80 (35.71)	43 (24.29)		1.00 (0.76; 1.31)
Obesity	62 (27.68)	58 (32.77)		
Bone mass				
Normal	122 (60.10)	89 (54.60)		
Osteopenia	48 (23.65)	65 (39.88)	0.000	1.49 (1.10; 2.02)
Osteoporosis	33 (16.26)	9 (5.52)		0.43 (0.21; 0.86)

**Table 2 jfmk-09-00012-t002:** Anthropometric variables of postmenopausal women.

Variable	Physical Activity		*p* Value	PR (CI 95% LCI; UCI)
	Active N (%)	Sedentary N (%)		
Waist-to-height ratio (WHtR)				

Low risk	22 (14.97)	10 (7.52)	0.050	1.08 (1.00; 1.18)
High risk	125 (85.03)	123 (92.48)		
Bone mass				
Normal	33 (22.92)	46 (35.11)		
Osteopenia	63 (43.75)	39 (29.77)	0.027	0.69 (0.53; 0.91)
Osteoporosis	48 (33.33)	46 (35.11)		0.84 (0.64; 1.10)

**Table 3 jfmk-09-00012-t003:** Anthropometric variables of physiological, premature, and early postmenopausal women.

	Physiological			Premature			Precocious		
Variable	Physical Activity		*p* Value	Physical Activity		*p* Value	Physical Activity		*p* Value
	**Active N (%)**	**Sedentary N (%)**		**Active N (%)**	**Sedentary N (%)**		**Active N (%)**	**Sedentary N (%)**	
Waist Circumference (WC)			0.407			0.032			0.519
Risk-free	16 (20.00)	10 (13.70)		6 (25.00)	0 (0.00)		7 (17.07)	5 (13.16)	
High risk	38 (47.50)	42 (57.53)		12 (50.00)	17 (77.27)		25 (60.98)	20 (52.63)	
Very high risk	26 (32.50)	21 (28.77)		6 (25.00)	5 (22.73)		9 (21.95)	13 (34.21)	

**Table 4 jfmk-09-00012-t004:** Prevalence ratio adjusted for age, race, and skin color.

Variable	PR (CI 95% LCI; UCI) *	PR (CI 95% LCI; UCI) **	PR (CI 95% LCI; UCI) ***
Overall Sample			
Waist Circumference (WC)			
Very high risk	1.12 (0.85; 1.32)	1.04 (1.01; 1.24)	1.12 (0.94; 1.15)
Did not go through menopause			
Bone mass			
Osteopenia	1.52 (1.23; 1.78)	1.36 (1.09; 1.62)	1.52 (1.23; 1.78)
Have already gone through menopause			
Waist Circumference (WC)			
Very high risk	1.29 (1.03; 1.55)	1.28 (1.03; 1.55)	1.29 (1.04; 1.56)
Waist-to-height ratio (WHtR)	1.60 (1.00; 2.20)	1.64 (1.03; 2.23)	1.65 (1.04; 2.25)

Adjusted for age; * Adjusted by Race/Color; ** Adjusted for age and race/color ***.

## Data Availability

The data presented in the present study may be available upon request.

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
