# Peer review of "Variations in Postmenopausal Body Composition: A Cross-Sectional Comparison between Physical Activity Practitioners and Sedentary Individuals"

_jfmk, 2023, doi:10.3390/jfmk9010012_

Round 1

Reviewer 1 Report

Comments and Suggestions for Authors

The work is of interest but major revision is needed.

There are two many details in the tables, too many numbers, so that the readers can not easily see the main results. 

Improvements may be for example: rather than separating in three BMI groups it could be more informative, and more visually, to see histograms for patients with and without phycical activity.

Some of the details can be presented in appendix and include in the text only main results as shorter more compact tables. For example in table 1, where there is no significant effects, show only the p-values wheres histograms or details of the groups can be in the appendix. For other results where there are significant effects, highlight those results as histograms for active vs sedant as figures in th text. 

All tables have to be on the same side, not split over two pages. 

Furthermore, it should be made explicitly described what the % is percentage off, thus add in 100% or describe this in the table labels.

In some of tables the lines of the tables fail to be on the right posistion.

There are errors in the table such as: 123 940.59)

The statistics in Table 5 is not clear. Where can it be seen that Waist Circumference represented a statistically. Furthermore, the three starts are missing below the table.

Table labels in table 6 is not in English.

With all these data at hand, is a two-group comparisons of Active vs Sedantary all you can consider, or did you consider that to make it fit into a simple two-groups statistics such as t-test?

Is there more information on the physical activity, such as type of activity, quantitative observations etc that can be utilized so that you could in stead run multivariate data analysis with numerous input information on the physical activity and all the responses as out put. If so I would recomend Partial Least Squares regression as a multivariate analysis.

Based on all this I recomend major revision to utilise these results far better and give a better communication to the readers.

Author Response

Dear editor, please find the response in the attached opinion.

Regards,

Camila Damasceno

Reviewer 2 Report

Comments and Suggestions for Authors

This study aims to clarify the relationship between physical activity and body composition in postmenopausal women. Defining the qualitative and quantitative effects of physical activity, including lifestyle, concerning body composition is very important for middle-aged and older women and women in transition to menopause from the perspective of improving quality of life and preventing chronic diseases. Therefore, I recognize the usefulness of this study.

As for the statistical methods, I understand that all items have been thoroughly reviewed and implemented.

Major Comments

In this study, the criteria for selecting and excluding research subjects are clearly stated in the "2.2Participants section". It is essential to clarify the selection criteria for research subjects to provide the study's rationale. However, the requirements for setting the sample size of the number of research subjects in this study need to be indicated. The criteria for setting the number of research subjects should be clarified at the beginning of every study. It is also essential to tell the requirements for developing the sample size in the research paper to show the research process. Adding an explanation of the criteria for setting the research subjects (sample size criteria) in the methodology would make the meaning of this study more straightforward. Please consider this.

The chronological flow in this study should be presented more transparently and understandably. For example, starting with recruiting research subjects, the number of recruits, the number of subjects studied, the number of issues excluded, etc., should be indicated. If possible, it would be helpful to present the flow chart of the study. Please consider this.

Please clarify the criteria for Active and Sedentary as the selection criteria; the term Aerobic exercise is mentioned, but no clear threshold is given. This selection criterion is essential for this study. Please consider them.

A comprehensive list of the attributes of study subjects (age, height, weight, BMI, measurements, etc.) should be presented at the beginning of the study's results section. Please consider this.

Author Response

(The authors gave the same response as above.)

Round 2

Reviewer 1 Report

Comments and Suggestions for Authors

The manuscript has improved considerably and is not far easier to read and understand. 

I still see the need for major improvements. 

Here are my suggestions:

All tables

All tables should be self-explaining. Therefor, explain in table text or below the tables, all shorts, such as: prevalence ratio (PR), confidence interval (CI), WHtR, WC, FMI etc should be defined

The definition of the groups is not clear. For example what are the definition: Normal, Overweight and Obesity etc etc for the other characters.

It should be more clear how the statistical test was performed. 

It is more difficult to understand what the CIs are as there are more than one CI for each charactre. Be explisit. Are tests performed within the categories, or across. 

Explain what the CI are: the difference: active - sedant or the opposite? subtraction or ratio: active/sedant or sedant/active. For example, for BMI, there are two CIs and three groups, What are the CIs ? 

The table must be corrected as the lines are not placed correctly.

It would be useful to know which variables were analysed by t-test and which were analysed  Mann-Whitney test. It would be useful to see the distribution in supplementary material. I do not understand why normalisation was not applied for example by log transformation. Did you investigate the data after normalsation? 

In general, it is very difficult to state negative results. If a bad experiment is conducted or bad statistics are conducted, one may not observe true effects. Thus false negative results are major problems for any experiments. In particular for this reason here, it is very important to be completely explicit in all steps that are made, and for example ensure that suitable preprocessing is considered.

Page 4. line 137
The word "most" refer to the first variable (age) but not to the rest. Suggestion: Most of the women were over 45 years of age (65.91%), the ratio of brown to white (42.06%) ......

Page 6, line 164
What is 1.18? In table 2, 1.18 appears to be the upper boundary of the confidence intervall

Page 6, Line 168-170 and Table 3
There is a need for better explanation in the text. For example: "The postmenopausal women are divided in three groups: physiological menopausal, premature menopausal, and early postmenopausal women. Only the  premature menopausal showed sidnificant differences in WC"

Use these fully explanations (physiological menopausal, premature menopausal, and early postmenopausal women) also in the table text.

Page 7, line 4-6
The text is: "Thus, it is understood that for this variable, women at very high risk, according to waist circumference, corrected for age and age x race/color, are 1.12 times more likely to be at very high risk for the group of women who do not practice physical activity."

Explain how the correction is made

Pate 7 line 12-13
The text says "For women who did not go through menopause, osteopenia adjusted for age (PR = 1.52), race/color (PR = 1.36), and age vs. race/color (PR = 1.52) showed a significant difference, Table 4". Explain the statistics.

Discussion

Page 8 line 37-38
This sentence is incomplete: "In this study, it was also observed that women adhered to the practice of physical activity." 

Main comment:
Separating the individuals into two groups "active" and "sedentary" is very rough. Care should therefore be taken in the conclusion. I would consider the results that are presented in this manuscript as indications, and need for deeper results than to state a strong conclusion. The conclusions do have formulations of this aspect, but I suggest to make that conclusion stronger, in particular referring to the rough separation in only two groups. 

I would think that more information could be extracted from this comprehensive data set. Maybe it is a reduction of the total information to separate only in two groups as is here done. Maybe there could be a following up publication that goes deeper into these data.

Author Response

Dear , here is the revised article.

Regards,

Round 3

Reviewer 1 Report

Comments and Suggestions for Authors

The manuscript have improved and I only have minor comments:

Page 8 line 11. One star is missing

Page 9 Kline 27. Instead of : "The other results of the study are presented in suppl material"

Could you say: More detailed results of the study are presented in suppl material". And should some of this be mentioned in the text?